# Toxicity of *Kadsura coccinea* (Lem.) A. C. Sm. Essential Oil to the Bed Bug, *Cimex lectularius* L. (Hemiptera: Cimicidae)

**DOI:** 10.3390/insects10060162

**Published:** 2019-06-07

**Authors:** Junaid U. Rehman, Mei Wang, Yupei Yang, Yongbei Liu, Bin Li, Yan Qin, Wei Wang, Amar G. Chittiboyina, Ikhlas A. Khan

**Affiliations:** 1National Center for Natural Products Research, The University of Mississippi, University, MS 38677, USA; jurehman@olemiss.edu (J.U.R.); meiwang@olemiss.edu (M.W.); amar@olemiss.edu (A.G.C.); 2TCM and Ethnomedicine Innovation & Development International Laboratory, Innovative Drug Research Institute, School of Pharmacy, Hunan University of Chinese Medicine, Changsha 410208, China; yangyupei24@163.com (Y.Y.); ybliu2018@163.com (Y.L.); libin_hucm@hotmail.com (B.L.); 0623_sister@163.com (Y.Q.); 3Department of BioMolecular Sciences, Division of Pharmacognosy, School of Pharmacy, The University of Mississippi, University, MS 38677, USA

**Keywords:** bed bug, *Kadsura coccinea*, topical toxicity, β-caryophyllene

## Abstract

*Kadsura coccinea* (Lem.) A.C. Smith is an evergreen, woody climbing plant that is widely distributed throughout southwest mainland China. Extracts of this plant are used in traditional Chinese medicine (TCM) for the treatment of various diseases, like cancer and dermatosis, and as an anodyne to relieve pain, while the leaves are used to treat eczema. In the current study, the toxicity of essential oil from its stem (EOKC) was studied against two strains of bed bugs (*Cimex lectularius*). Essential oil from the plant was obtained by hydrodistillation and analyzed by GC/MS. The major compound identified was β-caryophyllene (24.73%), followed by caryophyllene oxide (5.91%), α-humulene (3.48%), and β-pinene (2.54%). Preliminary screening was performed by topically delivering a 1 µL droplet of the treatments dissolved in acetone. At 24 h after treatment, the EOKC induced mortality rates of 61.9% and 66.7% in the Bayonne and Ft. Dix strains, respectively, at 100 µg/bug. Four major compounds—β-caryophyllene, caryophyllene oxide, α-humulene, and β-pinene—were selected based on their availability and were subjected to topical, residual, and fumigation methods. When applied topically, only β-caryophyllene induced high toxicity in both strains. None of the selected compounds induced significant toxicity in the residual and fumigation methods.

## 1. Introduction

*Cimex lectularius* (Lem.) A.C. Smith, popularly known as the bed bug, is a hematophagous pest that feeds on human blood. Its feeding behavior makes it a difficult pest to control. It typically feeds at night, whereas, during the day, it hides in the folds of furniture and inside cracks and crevices in both wooden and non-wooden structures [1,2]. It is small to medium-sized (4–12 mm), ovate, dorsoventrally flattened, and of brownish coloration [3,4,5,6]. Insecticides, especially pyrethroids, remain its main control measure, but bed bugs have developed a resistance to these compounds over time [7,8,9]. As bed bugs are an urban pest, the use of chemical insecticides, especially those with longer residues, to control them predispose humans to the risk of chemical exposure [6]. In recent decades, the bed bug population has resurged and become a major concern in the United States and other countries [6,10]. However, despite their pest status, a one-time application of bed bug treatment can minimize many risks affiliated with human–insecticide exposure. Natural products, like pyrethrum from the chrysanthemum flower [11], have been a source of new molecules that are a major base of pyrethroids pesticides. Plants use a diverse range of compounds in their defense against pests, which may have activities ranging from repellency to deterrence to lethality, and they generally leave no toxic residues. Exploration of such compounds from plants can help to design an effective bed bug control strategy. 

In this study, we explored the toxicity of the essential oil of stems from *Kadsura coccinea* (Lem.) A.C. Smith (EOKC) against the bed bug *Cimex lectularius*. *K. coccinea* is an evergreen climbing plant that is widely distributed throughout southwest mainland China. The dried roots and stems of this plant are known as “Heilaohu” in Chinese and are used by Chinese people to treat rheumatoid arthritis [12]. The stems of *K. coccinea* are also used for the treatment of gastroenteric disorders, duodenal ulcers, and gynecological problems in traditional Chinese medicine (TCM) [13]. *K. coccinea* is a rich source of lignans and triterpenoids, which have been reported to have several different bioactivities, including anti-inflammatory and neuroprotective effects [14]. Despite the medical importance of *Kadsura*, the essential oils from its other species have previously been reported to possess pesticidal toxicity. Specifically, *Kadsura heteroclite* shows fumigant activity against *Sitophilus zeamais* [15], *Kadsura longipedunculata* induces trypanocidal and funigicidal activity [16], and *Kadsura heteroclita* is larvicidal against *Aedes aegypti, Anopheles stephensi*, and *Culex quinquefasciatus* [17]. Essential oils are known to act as “pesticides” but are generally safer in terms of human toxicity against synthetic pesticides [11,18,19,20]. As no insecticidal data have been previously available on EOKC, we investigated the topical/contact toxicity properties of EOKC, while its major components were further investigated using topical, residual, and fumigant methods against two strains of bed bug: Bayonne (insecticide resistant) and Ft. Dix (susceptible).

## 2. Materials and Methodology

### 2.1. Materials

*Kadsura coccinea* was collected from HuaiHua City in Hunan Province, China. The plant was identified by Prof. Wei Wang (Director, TCM and Ethnomedicine Innovation and Development Laboratory, China). It was deposited at the TCM and Ethnomedicine Innovation & Development International Laboratory, Innovative Drug Research Institute, School of Pharmacy, Hunan University of Chinese Medicine, Changsha, China.

### 2.2. Sample Preparation

Twenty kilograms of powdered *K. coccinea* (dried) was divided into eight parts, with each part (2.5 kg) soaked in 3.5 L of water, subjected to hydrodistillation. The essential oil part was collected, dried over anhydrous sodium sulfate, and stored in an amber bottle.

### 2.3. GC/MS Analysis of K. Coccinea Essential Oil

Ten microliters of *K. coccinea* essential oil was dissolved in 1 mL of methylene chloride. GC/MS analysis was performed using an Agilent 7890A GC system fitted with a 7693 autosampler (Agilent Technologies, Santa Clara, CA, USA). The detection system used was an Agilent 5975C quadrupole mass spectrometer. The column used was a (5%-phenyl)-methylpolysiloxane (DB-5MS) capillary column (30 m x 0.25 mm I.D., 0.25 µm film thickness) from J&W Scientific (Folsom, CA, USA).

The oven temperature program conditions were as follows: 45 °C for 2 min, an increase to 110 °C at a rate of 2 °C/min, and then to 160 °C at a rate of 1 °C/min. The post-run was conducted at 280 °C for 5 min. The inlet temperature was 260 °C. The split injection was made with a split ratio of 50:1. Helium was used as the carrier gas at a flow rate of 1 mL/min.

The MS transfer line and MS source temperatures were 150 °C and 230 °C, respectively. Electron ionization mass spectra were recorded at 70 eV. Agilent MSD Chemstation (F.01.03) software (Santa Clara, CA, USA) was used for data acquisition and interpretation. 

Four major compounds, namely β-pinene, β-caryophyllene, α-humulene, and caryophyllene oxide, were identified by comparison of the spectra with databases (Wiley and NIST) and reference standards purchased from commercial sources. 

### 2.4. Chemicals

β-caryophyllene (CAS# 87-44-5), β-pinene (CAS# 18172-67-3), caryophyllene oxide (CAS# 1139-30-6), and α-humulene (CAS# 6753-98-6) were purchased from Sigma Aldrich (St. Louis, MO, USA). Certified acetone (Fisher, Suwanee, GA, USA) was used as a negative control. 

The rearing of bed bugs and toxicity testing of the test material followed the methodology given in our previous article [2]. 

### 2.5. Rearing of Bed Bugs

Two strains of bed bug (Bayonne “Insecticide-resistant” and Ft. Dix “Susceptible”) were provided by Dr. Changlu Wang, Department of Entomology, Rutgers University, New Brunswick, NJ, USA and were reared at the NCNPR since December, 2017. The active colony of bed bugs was maintained in 16-oz clear glass jars. Each jar had folded paper strips as harborages, and the mouth of the jar was covered with fine mesh cloth [21]. Jars were kept in a growth chamber at 26 °C and 60% relative humidity with a photoperiod of 13:11 h light/dark. Bed bugs were fed on defibrinated rabbit blood (Hemostat, Dixon, CA, USA) weekly [22]. Blood feeders (CG-1836-75 ChemGlass, Vineland, NJ, USA) were attached to a water bath, using latex tubing, and a piece of parafilm “M” membrane was stretched across the bottom of the feeder. Five milliliters of defibrinated rabbit blood (Hemostat) was placed in the hollow center of the glassware and pooled on the Parafilm. The Parafilm provided a barrier for the bed bugs to feed through. The water bath was set to circulate and warm the blood to 37 °C. The bed bugs dropped down after feeding and the jars were removed. Adult bed bugs of mixed age and sex (8–10 days post feeding) were used in all of the experiments. 

### 2.6. Topical Bioassay

This insecticidal study on the essential oil of *K. coccinea* “EOKC” and its compounds was conducted according to the procedure mentioned by Romero et al. [23]. Adult bed bugs of mixed age and sex were separated, placed in Petri dishes, and anesthetized with CO_2_. One microliter of a treatment solution in acetone was delivered onto the dorsal surface of the anesthetized bugs’ abdomens with a hand-held repeating dispenser (Hamilton Co., Reno, NV, USA), equipped with a 50 μL glass syringe (Hamilton Company, Reno, NV, USA) (Figure 1a). The control bed bugs received 1 μL of acetone alone. After treatment, bed bugs were transferred into 20 mL clear glass vials with paper strips. Bed bugs were kept in the growth chamber during this experiment. Bed bug mortality was recorded at 1, 2, 3, 5, and 7 days after treatment. Bed bugs were considered dead if no body part moved after touching the body with a needle. Tests were replicated three times with 15 bed bugs (mixed sex) per replication. Three doses of 25, 50, and 100 µg EOKC per bug were tested. Compounds were tested at 12.5, 25, 50, and 100 µg per bug. Deltamethrin (2.4 ng/bed bug) was used as a positive control. 

### 2.7. Residual Bioassay

A residual study of β-pinene, β-caryophyllene, α-humulene, and caryophyllene oxide was conducted with the no-choice method, using filter paper in the Petri dish described by Campbell and Miller [21] with slight modifications. A filter paper disc of 20 cm^2^ (Whatman 1) was treated with a 100 μL aliquot of each compound (100 and 300 μg/cm^2^) using a pipette and allowed to dry for 10 min. Dried, treated filter papers were then placed in the Petri dish (50 mm × 9 mm, FALCON, Franklin Lakes, NJ, USA) (Figure 1b). Five holes were drilled (1/16 of an inch drill bit) in the lid for aeration. Control treatments received only the carrier, acetone. Ten adult bugs of mixed sex were released on the filter paper and mortality was recorded for seven days, as mentioned previously. There were three replications, with 10 bugs per replication. Deltamethrin (6 µg/cm^2^) was used as a positive control.

### 2.8. Fumigation Bioassay

To evaluate the vapor toxicity of β-pinene, β-caryophyllene, α-humulene, and caryophyllene oxide, the bed bugs were exposed to vapor in 125-mL clear glass jars. A small piece of paper was placed at the bottom of the jar to provide a harbor for the bed bugs to rest on during the volatile tests. Bed bugs were introduced into the jars 2–4 h before treatment to acclimatize. A treatment solution (250 μg EOKC/125 mL air) in a 2-μL acetone aliquot was deposited directly onto the internal surface of the bottle side approximately 4 cm from the bottom, using a 50-μL gas-tight syringe (Hamilton Company) attached to a PB600 (Hamilton Company) repeating dispenser (Figure 1c). Jars were sealed immediately with a screw cap, followed by sealing with Parafilm “M”, and they were kept in the growth chamber. Mortality was recorded for 24 h after the treatment. Dichlorvos (1 µg/125 mL of air) was used as a positive control.

Mortality data from the bioassays were subjected to analysis of variance (ANOVA) using SAS^®^ 9.3 and JMP^®^ 10.0 [24]. The mortality rate in the essential oil was corrected by Abbott’s formula.

## 3. Results and Discussion

### 3.1. GC/MS Analysis

β-caryophyllene (24.73%) was identified as the major compound in *K. coccinea*, followed by caryophyllene oxide (5.91%), α-humulene (3.49%), and β-pinene (2.54%). Identity of these compounds was confirmed upon comparing with the respective reference compounds in terms of retention time, mass and mass fragmentation pattern (Figure 2).

### 3.2. Toxicity Results

The essential oil-induced mortality in both strains was observed to be dose-dependent. Of the three topically applied doses, EOKC induced significantly higher mortality at 100 µg/bed bug than at 50 and 25 µg in both strains of bed bug (Table 1). No statistical difference *F* (6.48 = 2.46, *p* = 0.037) was observed in the first three days, but the difference was more pronounced on days 5 and 7, especially at the highest dose (100 µg). The highest dose caused 61.9 ± 2.38% mortality in the Bayonne strain and 66.7% ± 2.38% in the Ft. Dix strain (24 h after treatment). Mortality reached 90.5% ± 2.38% in the Ft. Dix strain on day 5, but remained the same in the Bayonne strain. At 50 µg, 7.1% ± 0.00% mortality was recorded in the Bayonne strain 24 h after treatment and reached 33.3% ± 6.30% on day 7. In Ft. Dix., it was 11.9% ± 6.30% (24 h after treatment) and increased to 47.6% ± 2.38% on day 7.

Four major compounds—β-caryophyllene, (−)-caryophyllene oxide, α-humulene, and β-pinene—were selected based on their availability. Of the major compounds, only β-caryophyllene induced high toxicity and only at the highest dose of 100 µg when applied topically (Figure 3). In both strains, there was no difference between the percent mortality induced by β-caryophyllene (at 100 µg) from day 3 onward. It produced 83.3% ± 3.33% mortality (24 h after treatment) in the Bayonne strain, which increased to 86.7% ± 3.33% on day 7, while it was 60.0% ± 0.00% in Ft. Dix, which reached 83.3% ± 6.67% on day 7. At lower doses, the mortality recorded in the Bayonne strain was 20.0% ± 5.77% (50 µg), 13.3% ± 3.33% (25 µg), and 20.0% ± 0.00% (12.5 µg) at days 3, 5, and 7, respectively. In the Ft. Dix strain, mortality was 33.3% ± 8.82% (50 µg), 20.0% ± 5.77% (25 µg), and 20.0% ± 0.00% (12.5 µg) at days 3, 5, and 7, respectively. The mortality rates due to caryophyllene oxide (Figure 4), α-humulene (Figure 5), and β-pinene (Figure 6) remained very low (less than 40% on day 7) in both strains, even at the highest dose. Caryophyllene oxide induced mortality rates of 36.7% ± 3.33% (Bayonne) and 40.0 ± 15.28% (Ft Dix); α-humulene was associated with rates of 30.0% ± 5.77% (Bayonne) and 43.3 ± 3.33% (Ft. Dix); and β-pinene led to rates of 40.0% ± 0.00% (Bayonne) and 30.0% ± 0.00% (Ft. Dix) on day 7 after treatment at 100 µg.

The residual toxicity test revealed that none of the four compounds possessed toxicity against the bed bug as very low mortality rates were observed (Table 2). The highest mortality rates of 33.3% ± 12.01% (Bayonne) and 43.3% ± 8.81% (Ft. Dix) were observed with β-caryophyllene at 300 µg/cm^2^ on day 7.

Similarly, the four compounds were very weak fumigants against bed bugs (Figure 7). The highest mortality rate of 13.3% ± 3.33% (Ft. Dix) was produced by α-humulene in 250 µg of air. 

## 4. Conclusions

In summary, the topical assay results showed that exposure to EOKC at the highest dose led to significant mortality rates in both strains of bed bug investigated. Four major components of EOKC were studied using three different delivery methods: Direct spray (topical), surface deposit (residual), and fumigation (vapor toxicity). It should be kept in mind that a consumer has two expectations when using an insecticide: One is that it should kill pests by direct spray and the second is that it should provide control/protection for a certain length of time. If the target pest is hiding in cracks and the direct delivery of insecticide is not possible, then providing the chemical in a gaseous form can be highly effective. For a pest like bed bugs, a combination of the mentioned strategies is required. Our aim was to profile the toxicities of natural products using the three above-mentioned delivery methods to determine the specific potential of the test material. None of the four investigated compounds—β-caryophyllene, β-pinene, caryophyllene oxide, and α-humulene—were effective as surface deposits or vapors. In addition, of the selected components, only β-caryophyllene showed strong contact toxicity when applied topically, while none of the other compounds possessed topical, residual, or fumigant toxicity against bed bugs. In the past, researchers have reported [18] that β-caryophyllene, β-pinene [25,26], caryophyllene oxide, and α-humulene have contact, fumigant, and residual toxicity against various pests of stored grains, crops, and vector-borne diseases. However, only β-caryophyllene induced toxicity in adult bed bugs. As mentioned above, the delivery method and the penetration of the compound into the body of a bed bug will determine its effectiveness. Therefore, β-caryophyllene has mild topical applicability, while remaining non-toxic as a surface deposit and vapor. As such, correlating the contact toxicity of β-caryophyllene with the toxicity of EOKC is not justified because of the complexity of the interactions among the compounds in the oil. However, we aim to investigate the other minor compounds and their mixtures further in a follow-up study. To the best of our knowledge, this study is the first report of EOKC and its major compounds’ toxicity against both strains of bed bugs.

## Figures and Tables

**Figure 1 insects-10-00162-f001:**
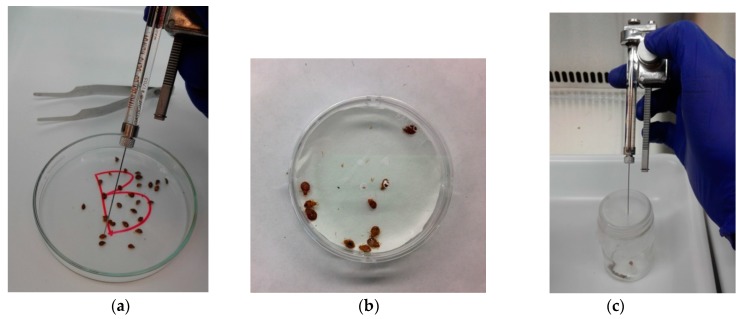
(**a**) Topical bioassay. (**b**) Residual bioassay. (**c**) Fumigant bioassay.

**Figure 2 insects-10-00162-f002:**
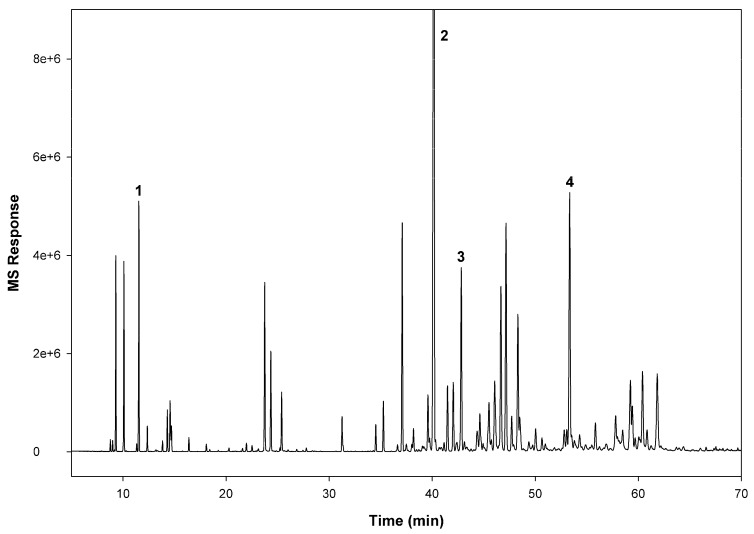
Total ion chromatogram of *Kadsura coccinea* essential oil. Compound identification: (**1**) β-pinene; (**2**) β-caryophyllene; (**3**) α-humulene; and (**4**) caryophyllene oxide.

**Figure 3 insects-10-00162-f003:**
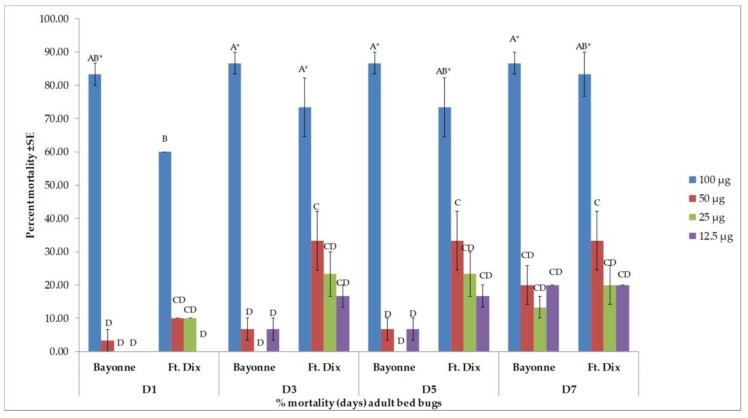
Mean (±SE) mortality produced by topical application of β-caryophyllene on two strains of bed bug. Levels not connected by the same letter are significantly different. There were three replications per treatment with 10 bed bugs per replication (Three-Factor ANOVA, *p* < 0.05, mean separated by Tukey’s HSD Test; JMP^®^ 10.0). Deltamethrin (2.4 ng, positive control) had killed 36.6% ± 3.33% (Bayonne) and 100% (Ft. Dix) of bed bugs at 24 h post treatment. The mortality rate of the Bayonne strain increased to 43.3% ± 8.82% for days 3–7. * The highest dose (100 µg) induced statistically higher mortality *F* (3.64 = 18.39, *p ≤* 0.0001) than the lower doses in both strains.

**Figure 4 insects-10-00162-f004:**
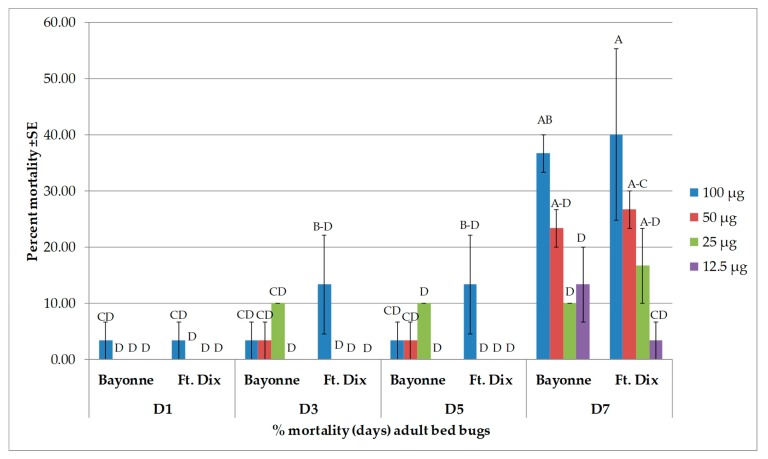
Mean (±SE) mortality produced by topical application of caryophyllene oxide on two strains of bed bug. Levels not connected by the same letter are significantly different. There were three replications per treatment with 10 bed bugs per replication (Three-Factor ANOVA, *p* < 0.05, mean separated by Tukey’s HSD Test; JMP^®^ 10.0). Deltamethrin (2.4 ng, positive control) had killed 36.6% ± 3.33% (Bayonne) and 100% (Ft. Dix) by 24 h post treatment. The mortality rate in the Bayonne strain increased to 43.3% ± 8.82% from days 3 to 7.

**Figure 5 insects-10-00162-f005:**
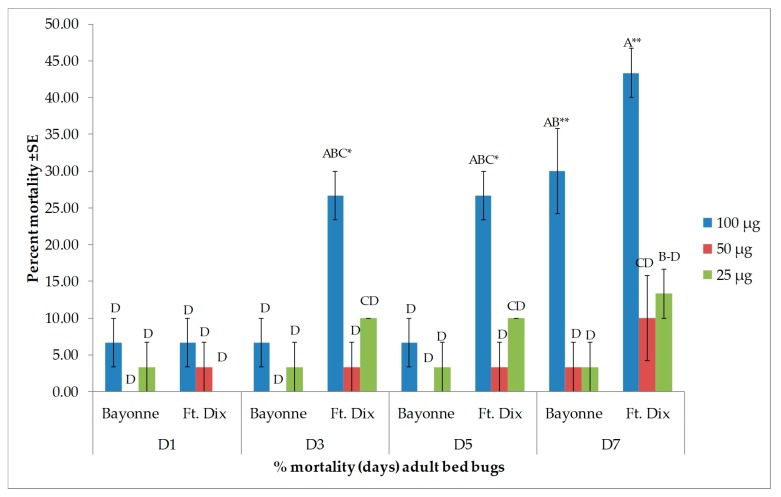
Mean (±SE) mortality produced by topical application of α-humulene on two strains of bed bug. Levels not connected by the same letter are significantly different. There were three replications per treatment with 10 bed bugs per replication (Three-Factor ANOVA, *p* < 0.05, mean separated by Tukey’s HSD Test; JMP^®^ 10.0). Deltamethrin (2.4 ng, positive control) had killed 36.6% ± 3.33% (Bayonne) and 100% (Ft. Dix) at 24 h post treatment. The mortality rate in the Bayonne strain increased to 43.3% ± 8.82% from days 3 to 7. * The highest dose (100 µg) induced statistically higher mortality in the Ft. Dix strain (susceptible) when compared with the Bayonne strain (insecticide-resistant) *F* (2.48 = 5.04, *p* ≤ 0.0103) on days 3 and 5, ** while the difference became non-significant on day 7.

**Figure 6 insects-10-00162-f006:**
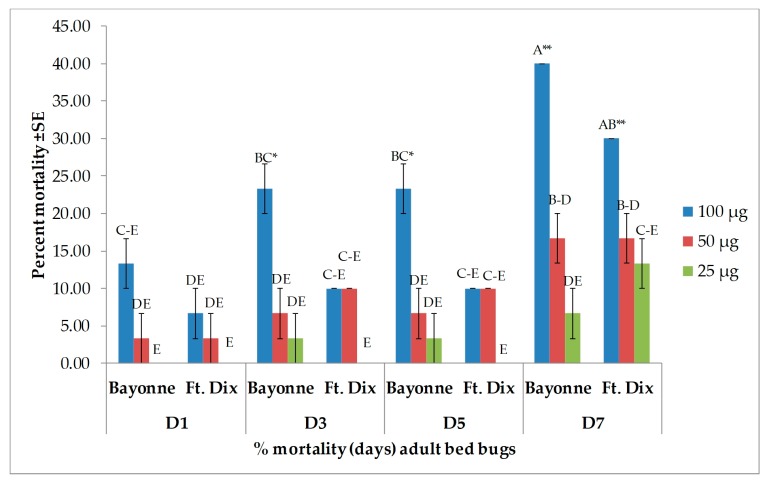
Mean (±SE) mortality produced by topical application of β-pinene on two strains of bed bug. Levels not connected by same letter are significantly different. There were three replications per treatment with 10 bed bugs per replication (Three-Factor ANOVA, *p* < 0.05, mean separated by Tukey’s HSD Test; JMP^®^ 10.0). Deltamethrin (2.4 ng, positive control) had killed 36.6% ± 3.33% (Bayonne) and 100% (Ft. Dix) at 24 h post treatment. The mortality in the Bayonne strain increased to 43.3% ± 8.82% from days 3 to 7. * Although the highest dose (100) induced higher mortality in the Bayonne strain than in the Ft. Dix strain on days 3, 5, and 7, ** this remained statistically non-significant.

**Figure 7 insects-10-00162-f007:**
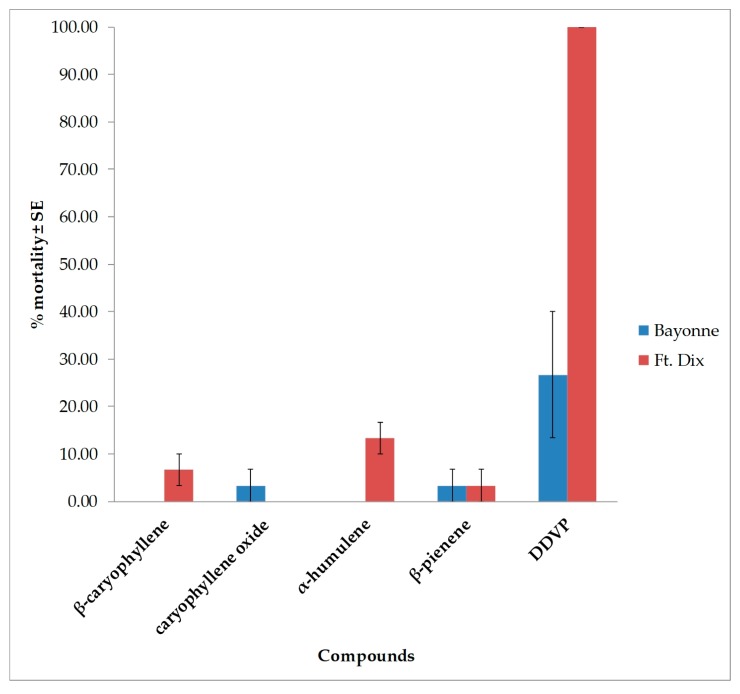
Mean (±SE) mortality produced by four compounds in two strains of bed bug in the vapor toxicity bioassay in 250 µg/125 mL of air. There were three replications per treatment with 10 bugs (adult mixed sex) per replication. The means and standard errors were calculated in f JMP^®^ 10.0.

**Table 1 insects-10-00162-t001:** Mean (±SE) mortality produced by essential oil of *Kadsura coccinea* “EOKC”, applied topically on two strains of bed bug.

Days after Treatment	Strain	Dose (µg/bed bug)		
		100	50	25
1	Bayonne	61.9 ± 2.38 ^B^	7.1 ± 0.00 ^EF^	0.0 ± 0.00 ^F^
	Ft. Dix	66.7 ± 2.38 ^B^	11.9 ± 6.30 ^DEF^	0.0 ± 0.00 ^F^
3	Bayonne	61.9 ± 2.38 ^B^	14.3 ± 4.12 ^DEF^	0.0 ± 0.00 ^F^
	Ft. Dix	66.7 ± 2.38 ^B^	28.6 ± 7.14 ^CDE^	4.8 ± 4.76 ^F^
5	Bayonne	61.9 ± 2.38 ^B^	16.7 ± 2.38 ^DEF^	7.1 ± 7.14 ^EF^
	Ft. Dix	90.5 ± 2.38 ^A^	28.6 ± 7.14 ^CDE^	4.8 ± 4.76 ^F^
7	Bayonne	61.9 ± 2.38 ^B^	33.3 ± 6.30 ^CD^	16.7 ± 8.58 ^DEF^
	Ft. Dix	90.5 ± 2.38 ^A^	47.6 ± 2.38 ^BC^	9.5 ± 2.38 ^EF^

Levels not connected by the same letter are significantly different. There were three replications per treatment with 15 bed bugs per replication (Three-Factor ANOVA, *p* < 0.05, mean separated by Tukey’s HSD Test; JMP^®^ 10.0). The mortality rate of the control bed bug of both strains, Bayonne and Ft. Dix, was 6.67%, and this was corrected by Abbot’s formula.

**Table 2 insects-10-00162-t002:** Mean (±SE) percent mortality produced by four major components of “EOKC” in two strains of bed bug, based on a residual toxicity bioassay at two different dose rates.

Strain	DAY	β-caryophyllene		caryophyllene oxide		α-humulene		β-pinene	
		100 *	300 *	100 *	300 *	100 *	300 *	100 *	300 *
**Bayonne**	1	0.0 ± 0.00	0.0 ± 0.00	0.0 ± 0.00	0.0 ± 0.00	0.0 ± 0.00	0.0 ± 0.00	0.0 ± 0.00	0.0 ± 0. 00
	3	0.0 ± 0.00	0.0 ± 0.00	0.0 ± 0.00	0.0 ± 0.00	0.0 ± 0.00	0.0 ± 0.00	0.0 ± 0.00	0.0 ± 0.00
	5	3.3 ± 3.33	30.0 ± 15.27	3.3 ± 3.33	26.7 ± 21.85	10.0 ± 0.00	20.0 ± 5.77	0.0 ± 0.00	3.3 ± 3.33
	7	6.7 ± 6.67	33.3 ± 12.01	3.3 ± 3.33	30.0 ± 20.81	13.3 ± 3.33	26.7 ± 8.81	0.0 ± 0.00	10.0 ± 3.33
**Ft. Dix**	1	0.0 ± 0.00	0.0 ± 0.00	0.0 ± 0.00	0.0 ± 0.00	0.0 ± 0.00	0.0 ± 0.00	0.0 ± 0.00	0.0 ± 0.00
	3	0.0 ± 0.00	0.0 ± 0.00	0.0 ± 0.00	0.0 ± 0.00	0.0 ± 0.00	0.0 ± 0.00	0.0 ± 0.00	0.0 ± 0.00
	5	13.3 ± 3.33	40.0 ± 5.77	3.3 ± 3.33	13.3 ± 8.81	13.3 ± 3.33	30.0 ± 5.77	3.3 ± 3.33	13.3 ± 6.67
	7	16.7 ± 3.33	43.3 ± 8.81	3.3 ± 3.33	16.7 ± 8.81	13.3 ± 3.33	36.7 ± 3.33	3.3 ± 3.33	16.7 ± 3.33

* Dose (µg/cm^2^), DAT: days after treatment. There were three replications per treatment with 10-bed bugs per replication. Deltamethrin (positive control) had killed 3.3% ± 3.33% (Bayonne) and 100% (Ft. Dix) of bed bugs at 24 h post treatment, and these percentage mortalities did not change until day 7.

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
