# Peer review of "Toxicity of *Kadsura coccinea* (Lem.) A. C. Sm. Essential Oil to the Bed Bug, *Cimex lectularius* L. (Hemiptera: Cimicidae)"

_insects, 2019, doi:10.3390/insects10060162_

Round 1
Reviewer 1 Report
The researchers tested the toxicity of essential oil extracts from a Chinese plant Kadsura coccinea, which has medicinal uses, against bed bugs. This team also tested four main components of the plant essential oil extract for toxicity against bed bugs topically, as a residual and as a vapor fumigant. Results showed that the essential oil induced mortality when applied topically. Other results showed that one component of the extract, beta caryophyllene, had significant topical toxicity (over 50%) =, whereas the other three compounds did not reach over 50%. No components showed significant mortality to bed bugs via residual toxicity or as a vapor toxicant. In the pursuit of lower risk and botanical compounds for use in pest management, and particularly bed bug management, this work identifies the potential for beta caryophyllene to be used as an active ingredient for bed bug control products.
In the abstract and introduction I, the reader, could not identify why the researchers were testing this extract against bed bugs, when it is commonly used as an herbal medicine. Only later in the discussion did I read that the components have insecticidal activity, but this statement is poorly referenced. This justification needs to be brought forward into the introduction and references made stronger (what about caryophyllene oxide and alpha-humulene?).
Methods: Bed bugs used in studies of insecticide efficacy are always standardized by life stage and feeding status. This was not mentioned in the article and if the researchers did not standardize the bed bugs used in these experiments, that could introduce variation that was not accounted for. If they did, it should be stated.
Also in Methods there are several articles referenced (Romero et al 2009, Montes et al 2002, Campbell and Miller 2015) that are not found in the list of references!
The US EPA “Guidance of Efficacy Testing for Pesticides Targeting Certain Invertebrate Pests” recommends that a minimum of five replicates of ten individuals per replicate be used (not three) and that specimens exposed to toxicant be transferred to clean containers after pesticide treatment because continuous exposure to a toxicant is not a realistic scenario in most cases. This was not done in any of the experiments reported in this manuscript.
Analysis: Although this is preliminary work, the majority of insecticide efficacy studies use a dose-response measurement and log-probit analysis to determine the LD50/LD90 or LT50/LT90. It would have been a better study if there were additional doses and determination of LD50 and LD90. I am also left wondering about control mortality in all experiments.
Author Response
Dear Reviewer,
We thank you for the helpful comments and valuable input. We also appreciate your efforts in the review of our manuscript. Our responses to the your suggestions/ comments is in the attached file. Specific changes to the manuscript were highlighted and attached as revised manuscript.

Reviewer 2 Report
Specific comments:
L15: The scientific name here is different from the title, make them consistent
L19: Change bed bug ‘Cimex lectularius’ to bed bug, Cimex lectularius
L22: Change 3.485 to 3.49%
L22: Add “,” before “and”. When presenting multiple independent subjects, it should always be “A, B, and C”. This issue took place in the whole manuscript, please check and revise them.
L33: Add “as” before bed bug
L41: “For the pest like, bed bug, one time application of treatment can minimize many risks affiliated to human-insecticide exposure.”
Many previous studies emphasized the importance of multiple follow-up treatments. Even for labor-intense thorough treatment, it’s unlikely that bed bugs can be eliminated by a single treatment; not to mention “green pesticides”, which is pretty mild with no residual effects compare to other tools.
L57: Revise “twos trains” to “two strains”
L66: Add “was” before “devided”
L67: Delete “its”
L70: “essential oil” should not be italic
L110: Delete “a” before treatment
L113: Delete “the” before 20 ml
L113: Change “having” to “with”
L118: Add space between 2.4ng. The space between number and unit has been omitted here and there throughout the manuscript, please check and revise
L118: Add “was” before used
L128: Add “was” before used
L131: Delete “of”
L137: Delete “it”
L146: Add % after 3.49
L156-160: Check the font of the standard errors; it looks different
L170-171: Same comment as above
L173-174: Add space before and after “±”
L173: change 20.00 to 20.0
L213: Revise “the adults of bed bug ‘C. lectularius’” to “bed bug adults”
L216: Change “in case of” to “in the case of”
L219: Change “To our best of knowledge” to “To the best of our knowledge”
L219: “this study is the first reporting on bed bug” Please specify. Also change reporting to report.
Overall comments:
Writing and organization of the manuscript should be significantly improved to be considered for publication. There are sentences that make little sense and do not justify themselves, for example L39, “bed bug is a zero tolerant pest that can not be tolerated.” In L35, “Bed bug is small to medium-sized (4-12 mm), ovate, dorsoventrally flattened and of brownish coloration.” The author cited 4 literatures for such a simple and obvious description; however, no previous work was cited at all to support the author’s statements in pesticide resistant, chemical exposure risks to human, and so on.
Almost half of the introduction part focused on the medical effects of K. coccinea, which had little to do which the author’s subject. Instead, the author should justify this study by emphasizing the following aspects:
1. The medical and economical importance of bed bug.
2. The challenges and difficulties of bed bug management.
3. Disadvantages of conventional pesticides: resistance & chemical exposure risks to human & environment.
4. Previous studies and practical applications of green pesticides for pest control, including bed bug.
5. The reason to choose K. coccinea as study object. Various beneficial medical effects were listed in the introduction, however none of those was correlated with insecticidal activity. Why did the author choose K. coccinea as the object of study, and what was the expectation and hypothesis?
In sum, the introduction should be presented in a more logical way which clearly justify the study background, objectives, and hypothesis. There are minor errors and typos all over the place; grammar, format, and writing of the whole manuscript need to be carefully revised and improved, include but not limited to my specific comments.
Author Response

(The authors gave the same response as above.)

Reviewer 3 Report
See attached.

Author Response
Dear Reviewer,
We thank you for the helpful comments and valuable input. We also appreciate your efforts in the review of our manuscript. Our responses to your comments are given in the attached file. Specific changes to the manuscript were made and attached as revised manuscript.

Round 2
Reviewer 1 Report
Only grammatical edits in summary, attached file.

Author Response
Dear Reviewer,
We thank you for your valuable time and input to refine this article. The manuscript has been updated as suggested. Dr. John F. Parcher has also reviewed the manuscript for English grammer correction and has been acknowledged for that in the acknowledgement section.
Regards

Reviewer 2 Report
I saw improvement in writing and organizing.
In response to reviewer 1, the author added L 113-114: "Bed bugs 8-10 days post feeding were used in the experiment." This does not indicate the life stages of the bed bug; please add life stage information. If bed bugs were assigned to each groups in a completely randomized manner, please state "mixed-age".
I agree that LD50 and LD90 are not essential for this particular study, given that the toxicity of tested essential oil was too low to get a meaningful LD50 value. However, I am also wondering about control mortality. The author should definitely add negative control mortality even though the mortality data has been adjusted by Abbott's formula. Usually you need a control mortality<10% to prove the validity of the experiment method.
Author Response
Dear Reviewer,
We
thank you for your valuable time and input to refine this article. The
manuscript has been updated as suggested. Dr. John F. Parcher has also
reviewed the manuscript for English grammer correction and has been
acknowledged for that in the acknowledgement section.
Regards
'
